# Mechanisms of Impact of *Alnus ferdinandi-coburgii* Odor Substances on Host Location of *Tomicus yunnanensis*

**DOI:** 10.3390/insects16060553

**Published:** 2025-05-23

**Authors:** Jingyi Bo, Wen Li, Xiangyi Li, Zongbo Li, Xiangzhong Mao, Bin Yang, Ning Zhao

**Affiliations:** 1College of Biological Science and Food Engineering, Southwest Forestry University, Kunming 650224, China; zjingbojingyi@163.com (J.B.); liwenaya@163.com (W.L.); lxylxylix@163.com (X.L.); xlsktw@swfu.edu.cn (X.M.); 2Key Laboratory of Forest Disaster Warning and Control of Yunnan Province, Southwest Forestry University, Kunming 650224, China; lizb@swfu.edu.cn

**Keywords:** chemosensory-related genes, *Tomicus yunnanensis*, host location, non-host, molecular docking

## Abstract

The control of *Tomicus yunnanensis*, a major pest of Yunnan pine, remains challenging due to the high cost and environmental impact of traditional methods. Mixed forests show promise in pest resistance, but the underlying mechanisms are unclear, potentially involving insect chemosensory proteins. Preliminary studies indicate that bioactive compounds from *Alnus ferdinandi-coburgii* (a key mixed-forest species) significantly disrupt *T. yunnanensis* development. This study investigates how these compounds interact with the pest’s chemosensory proteins, aiming to clarify mixed-forest resistance mechanisms and offer eco-friendly control strategies.

## 1. Introduction

The chemosensory system of insects is crucial for their survival and reproduction, enabling them to identify and analyze odorants, thereby distinguishing unique chemical signals in their living environment [1,2]. These signals are crucial for insects’ ecological adjustments, pinpointing host plants, selecting mates, engaging in reproductive activities, and carrying out essential physiological functions. The insect chemosensory system consists of a variety of chemosensory-related proteins that hierarchically process various signals collected by insects in the outside world [3]. The function of the chemosensory system of insects is primarily mediated by several types of chemosensory-related proteins: odorant-binding proteins (OBPs) and chemosensory proteins (CSPs), which are involved in the recognition and binding process of odors [4]; and three main families of chemosensory receptors, namely, the olfactory receptors (ORs) for olfaction, gustatory receptors (GRs) for taste and contact chemosensation, and ionotropic receptors (IRs) involved in olfaction, taste, temperature, and humidity sensation [5]. In addition, sensory neuron membrane proteins (SNMPs) also participate in this process [6]. These proteins work together to enable insects to effectively recognize and respond to chemical signals in the environment.

OBPs and CSPs are located in the lymphatic fluid of the insect antennal sensitizer and can accurately bind to external odorants and translocate them to the corresponding ORs, IRs, or GRs to initiate behavioral responses. Thus, the role of OBPs and CSPs is the first step in activating olfactory perception in insects [7,8,9]. OBPs are one of the major classes of proteins responsible for binding and transporting water-insoluble compounds through the sensory lymphatic fluid. OBPs are small, water-soluble proteins, and a typical OBP contains six conserved cysteine residues. The endohydrophobic pocket for lipophilic ligand binding is established by the interconnection of disulfide bonds. In addition, OBPs with more or less than six conserved cysteines have been identified and named Plus-C OBPs and Minus-C OBPs, respectively [10]. There is a group of OBPs that have been termed “atypical OBPs” characterized by 10 or more conserved cysteines, a long C-terminus, a conserved proline residue, and four or more disulfide bridges, which is recorded in several mosquito and locust species, suggesting that this group of genes may be recently evolved in these species. Typically, external odorants or volatile compounds pass through cuticular pores into the sensillum lymph, where they are bound and solubilized by OBPs. These OBP–odorant complexes are then transported through the sensillum lymph to potential receptors, enabling signal transduction [11]. However, the olfactory process does not end with receptor activation. To prevent olfactory fatigue and maintain sensitivity to new stimuli, odor molecules must be rapidly degraded and inactivated. This critical function is carried out by odorant-degrading enzymes (ODEs), which have been extensively studied since their initial characterization by Vogt et al. in the 1980s and 1990s [5,9,12]. ODEs are divided into various types, the most prominent of which are cytochrome P450 monooxygenases (CYPs), glutathione S-transferases (GSTs), and carboxylesterases (CCEs). CYP enzymes typically serve as the body’s first line of defense for metabolizing exogenous substances, responsible for the oxidation, peroxidation, and reduction of various endogenous and exogenous substrates [13]. GSTs can detoxify a wide range of plant molecules. Typically, different GSTs exhibit overlapping substrate specificity, but during diversification, certain GSTs may evolve specialized functions in metabolizing specific compounds [14]. CCEs are key enzymes in the nervous system, whose function in animals is to terminate nerve transmission by rapidly hydrolyzing the neurotransmitter acetylcholine at cholinergic synapses [13]. These enzymes are not only expressed in antennal tissues but also play diverse roles beyond odorant degradation, including detoxification of xenobiotics, which has led to their classification as detoxification enzymes [15,16,17,18]. In addition to the aforementioned enzymes, recent studies have suggested important roles of UDP-glycosyltransferases in insect chemosensory systems [19]. However, due to limited published data on the identification of UDP-glycosyltransferases in Coleopteran insects, this study does not include a relevant analysis of these enzymes. The interplay between OBPs and detoxification enzymes is well documented in the literature. For instance, cytochrome P450 enzymes, which are central to insecticide resistance, are also implicated in odorant degradation, highlighting the functional overlap between these systems [20,21]. Furthermore, studies have shown that ODEs can degrade excess odor molecules in the sensilla lymph and within cells, with different ODEs exhibiting specificity for odorants of similar chemical structures [18]. This intricate system underscores the evolutionary adaptation of insects to their chemical environments. It provides a foundation for exploring ODEs and OBPs as targets for insect control strategies, an idea that has been explored since the early 2000s [9,20,22].

*Tomicus yunnanensis* (*T. yunnanensis* Kirkendall and faccoli, 2008, Coleoptera, Curculionidae) is one of the main pests of *Pinus yunnanensis* (*P. yunnanensis*, Franch, 1899, Pinales, Pinaceae) with a long period of harm and a wide area of damage [23,24]. Currently, the main methods for controlling *T. yunnanensis* include manual cleaning of infested trees, chemical control, and lure killing, but these methods either consume a lot of manpower and resources or pollute the environment [25,26]. Mixed-forest resistance is a green and sustainable method of control which improves resistance to *T. yunnanensis* by increasing tree species diversity [27,28]. Studies have shown that the volatile odor substances of non-host plants can interfere with the host location of *T. yunnanensis* and affect its olfactory behavior [29]. *Alnus ferdinandi-coburgii* (*A. ferdinandi-coburgii* Schneid, 1917, Fagales, Betulaceae) is of great value in the study of mixed-forest resistance to *T. yunnanensis*. The volatile odor substances of *A. ferdinandi-coburgii* affect the ovarian development and egg-laying quantity of *T. yunnanensis*, among which 4′-ethylacetophenoneis is the main volatile substance in *A. ferdinandi-coburgii*, showing a significant repellent effect on *T. yunnanensis* [30,31].

In summary, the chemosensory system mediates the recognition and response of insects to volatile compounds in the environment through a variety of chemosensory-related proteins, and the identification of chemosensory-related proteins of *T. yunnanensis* is of great value. In the control of *T. yunnanensis*, the mixed-forest resistance method uses the volatile odor substances of non-host plants to interfere with the host location of the pest, showing good resistance effects. It is an environmentally friendly and effective control strategy, but the mechanisms by which the volatile substances of non-host plants interfere with the host location of pests are not yet clear. The integration of crosstalk studies between chemosensory-associated proteins and host or non-host volatile odorants is a viable current direction.

## 2. Materials and Methods

### 2.1. Insect and Tissue Collection

During the preliminary research phase, transcriptome data from the antennae, heads, legs, and residual bodies of male and female adults of *T. yunnanensis* were obtained [32]. The male and female adults of *T. yunnanensis* used in this experiment were collected from September 2020 to June 2021 from Jiulong Mountain Forest Farm in Zhanyi County, Qujing City, Yunnan Province, which is the same location where the insects for transcriptome data collection were gathered. Adult of *T. yunnanensis*, once collected, were subjected to a week of feeding in the laboratory (nourished with *P. yunnanensis* shoots) before being dissected for tissue collection. A total of 200 pairs of antennae, heads, legs, and residual bodies (with antennae, heads, and legs removed) of male and female adults of *T. yunnanensis* were collected, with three biological replicates for each. The collected samples were rapidly frozen with liquid nitrogen and stored in a refrigerator at −80 °C. *T. yunnanensis* treated with compounds were subjected to 8 h and 24 h in a 6 °C refrigerator. After treatment, the antennae and residual bodies (with antennae, heads, and legs removed) of male and female adults were subjected to liquid nitrogen freezing treatment and preserved in a refrigerator at −80 °C.

### 2.2. Identification of Chemosensory-Related Genes of T. yunnanensis

In order to identify the chemosensory-related genes (*OBP*, *CSP*, *OR*, *GR*, *IR*, and *SNMP*) of *T. yunnanensis*, this study obtained known amino acid sequences of *OBP*, *CSP*, *OR*, *GR*, *IR*, and *SNMP* from other homologous insects (*Tribolium castaneum*, *Ips typographus*, *Dendroctonus armandi*, *Dendroctonus ponderosae*, *Triatoma rubrofasciata*, *Dendroctonus valens, and Anoplophora glabripennis*) from the NCBI database (https://www.ncbi.nlm.nih.gov). The retrieved sequences were used to screen the unigenes of *T. yunnanensis*, followed by homology comparison searches using the TBtools software BLAST (V2.225). Through the comparison, candidate chemosensory-related genes were obtained, and then manual homology comparison searches were conducted using the Protein BLAST in the NCBI database (https://www.ncbi.nlm.nih.gov) to determine the chemosensory-related genes of *T. yunnanensis*. All identified open reading frames (ORFs) of chemosensory-related genes of *T. yunnanensis* were predicted using the ORF finder online tool (https://www.ncbi.nlm.nih.gov/orffinder/, accessed on 9 January 2021). The transmembrane domains of the identified ORs, GRs, and IRs of *T. yunnanensis* were predicted using the online software TMHMM 2.0 (http://www.cbs.dtu.dk/services/TMHMM/, accessed on 9 January 2021). Subsequently, the signal peptides of OBPs and CSPs were predicted using SignalP 4.0 (http://www.cbs.dtu.dk/services/SignalP/, accessed on 9 January 2021).

### 2.3. Sequence Analysis and Phylogenetic Tree Construction

After identification, the software RNA-Seq by Expectation-Maximization (RSEM 1.3.3) was used to perform quantitative analysis of gene expression levels. Subsequently, FPKM (Fragments Per Kilobase of transcript, per Million mapped reads) was used to assess the differences in expression levels of *OBP* genes in four different tissues of male and female adults of *T. yunnanensis*. To further visualize and interpret the gene expression patterns, TBtools 2.225 was utilized to generate heatmaps. In the set of trees, multiple sequence alignment was performed using the Muscle method in MEGA7.0, and the neighbor-joining (NJ) method in MEGA7.0 software was used to construct the phylogenetic tree of chemosensory-related proteins (Bootstrap = 1000) [33]. Evolview v2 was used to visualize the phylogenetic tree [34].

### 2.4. Influence of Non-Host Odorants on Chemosensory-Related Genes of T. yunnanensis

Using fumigation toxicity tests, 90 mL culture dishes (90 × 15 mm) were filled with 0, 9, 18, 54, 90, and 180 uL of the compound (α-pinene or 4′-ethylacetophenone) at compound concentrations of 0, 100, 200, 600, 1000, and 2000 ppm, respectively [35]. A total of 20 adult *T. yunnanensis* (10 females + 10 males) were placed into culture dishes lined with filter paper, and the treated dishes were placed in a 6 °C refrigerator for 24 h. After 24 h, the dishes were removed from the refrigerator and allowed to stand for 30 min before observing the survival of *T. yunnanensis* and calculating the mortality rate.

Thirty adult *T. yunnanensis* were placed into 90 mL culture dishes (90 × 15 mm) lined with filter paper. The compound (lethal concentration 50) was added to the top filter paper, and the dish was immediately sealed with parafilm. Each compound treatment was divided into 6 culture dishes. The treated dishes were placed in a 6 °C refrigerator for 8 and 24 h, and the survival of the beetles was observed under a dissecting microscope after treatment. When the mortality rate was between 40 and 50%, live beetles were selected for dissection, and the antennae and residual bodies were taken for subsequent experiments. Untreated beetles served as a blank control, and host compound (α-pinene) treatment served as a negative control for qRT-PCR analysis of the selected genes. In the selection of key genes for subsequent research, the OBP genes were prioritized based on their antennal-specific high expression patterns. The CYP genes were selected based on both their high tissue expression levels and their classification into the CYP4 and CYP6 subfamilies, which are closely associated with insect-mediated degradation of plant secondary metabolites [32]. Gene-specific primers for qRT-PCR were designed using Primer 7.0 (Appendix A). Each reaction contained a total volume of 20 μL, consisting of 10 μL of SYBR Green PCR Master Mix (TaKaRa, Japan), 0.8 μL of each primer (10 μM), 2 μL (20 ng) of cDNA template, and 6.4 μL of nuclease-free water. The β-actin gene was used as an endogenous control [36]. qPCR cycling parameters were as follows: 94 °C for 4 min followed by 40 cycles at 94 °C for 20 s and 60 °C for 30 s. Relative gene expression level was calculated using the Q-GENE statistical analysis package (according to the following formula: relative gene expression = 2^−ΔΔct^), and the up- and downregulated levels of gene expression in the treated group were analyzed using the untreated control group as a reference.

### 2.5. Molecular Docking and Molecular Dynamic Simulation

Protein three-dimensional modeling was conducted using the homology modeling server SWISS-MODEL (https://swissmodel.expasy.org/, accessed on 17 January 2022). The optimal protein for homology modeling was selected based on the similarity between the protein sequence and the template sequence, protein homology, and the presence or absence of ligands in the crystal structure for subsequent molecular docking analysis.

Based on the two active odor components from the preliminary compound treatment, namely the host compound α-pinene and the non-host compound 4′-ethylacetophenone, which serve as small molecular ligands for molecular docking, their three-dimensional structures were downloaded from PubChem (https://pubchem.ncbi.nlm.nih.gov/, accessed on 17 January 2022). The highly expressed *TyunOBPs* and *TyunCYPs* in the antennae and residual bodies, which are likely to have functions related to the compounds, were used as target proteins. Molecular docking was performed separately with the two compounds using AutoDock4.0. The docking pocket was designed based on a grid box set as the largest cube enclosing the protein, while AutoDock4.0 automatically identified the central binding pocket, and the top ranked binding mode was evaluated according to the docking score and visually analyzed by PyMOL (version 1.9.0) [37,38].

To further evaluate the binding degree and stability between compounds and proteins, 100 ns molecular dynamics (MD) simulations were performed on complexes formed by pairwise combinations of proteins TyunCYP4G2 and TyunOBP6 with 4′-Ethylacetophenone and α-Pinene. The MD simulations for each protein–ligand complex were conducted using Gromacs (version 4.6.3) software with the Amber99sb force field for proteins and GAFF2 force field for ligands [39]. The CHARMM-modified TIP3P water model was employed to solvate the protein–ligand systems within periodic boundary conditions using a 1.0 nm water box [40]. The Particle Mesh Ewald (PME) method was applied for calculating long-range electrostatic interactions [41]. The simulation trajectories were analyzed through multiple parameters including the root-mean-square deviation (RMSD), root-mean-square fluctuation (RMSF), radius of gyration (Rg), solvent-accessible surface area (SASA), and relative free energy distribution assessments of the protein–ligand complexes [42].

## 3. Results

### 3.1. Basic Information and Expression Patterns of Chemosensory-Related Genes

In this study, a total of 137 chemosensory-related genes were identified from the transcriptomes of different tissues of male and female adults of *T. yunnanensis*, including 32 *TyunOBPs*, 11 *TyunCSPs*, 36 *TyunORs*, 19 *TyunGRs*, 31 *TyunIRs*, and 8 *TyunSNMPs* (Appendix A). The sequence homology of these TyunOBPs with other Coleoptera insects ranges from 34.62% to 100%, all possessing complete open reading frames with lengths between 41 and 500 amino acids (aa). Among them, 23 TyunOBPs have signal peptides, while the remaining 9 sequences lack signal peptides. Out of the 11 TyunCSPs, 10 encode full-length protein sequences with predicted signal peptide sequences, with sequence homology ranging from 61.54% to 100%. The sequence homology of the 36 TyunORs with other Coleoptera insects varies from 28.22% to 96.76%, and all TyunORs contain 1–10 transmembrane domains (TMDs). The 19 candidate GRs all have complete open reading frames, with 6 TyunGRs having 4–8 TMDs and 13 sequences having 0–3 TMDs. The sequence homology of the 31 TyunIRs with other Coleoptera insects ranges from 53.41% to 96.6%, and the 8 encoded TyunSNMPs share 47.33–89.12% amino acid homology with SNMPs from other insects.

Based on the FPKM values, a gene expression map of chemosensory-related genes in various tissues of *T. yunnanensis* was established, revealing that the *TyunOBP* genes are most abundantly expressed in the antennae. The expression levels of *TyunOBP2*, *TyunOBP6*, *TyunOBP9*, *TyunOBP11*, and *TyunOBP50* in the antennae are significantly higher than in other tissue regions. Among these, *TyunOBP6* has the highest expression in both male and female antennae, with a notably higher expression in male antennae compared to female antennae (Figure 1).

### 3.2. Phylogenetic Analysis of Chemosensory-Related Proteins

Phylogenetic tree analysis of OBPs from Coleoptera species and *T. yunnanensis* indicates that TyunOBPs were divided into five families, including the Atypical, Minue-C, ABPII, Plus-C, and Classic families. The Plus-C OBP subfamily includes TyunOBP46, which clusters with Coleoptera sequences DponOBP2 and ItypOBP2Fix, with a bootstrap value of 100%. There were six TyunOBPs present in the Classic branch, which contains a large number of genes, including TyunOBP3 with DponOBP22, TyunOBP6 with DponOBP5, TyunOBP11 with DponOBP8, TyunOBP59 with Dpon34, and TyunOBP62 with Dpon33, all with bootstrap values of 100%. The Classic OBP subfamily is further classified as the “Antennae-binding protein II” family (ABPII), and two TyunOBPs (TyunOBP6, TyunOBP10) belong to the ABPII. There are nine TyunOBPs present in the Minue-C subfamily, including TyunOBP4, TyunOBP47, TyunOBP51, Tyunobp5, TyunOBP54, TyunOBP56, TyunOBP57, TyunOBP61, and TyunOBP64, which cluster together with DponOBP9, DponOBP11, DponOBP4, DponOBP7a, DponOBP7b, DponOBP19, DponOBP25, DponOBP28, DponOBP39, TcasOBP9, DponOBP36, ItypOBP15, ItypOBP7, and TcasOBP3 (Figure 2). Additionally, most TyunOBPs cluster with those of the *Dendroctonus ponderosae* on the same branch, with high bootstrap values, indicating a close phylogenetic relationship between the OBPs of these two bark beetles and suggesting that their proteins may have similar functions.

The phylogenetic tree constructed with CSP sequences from *T. yunnanensis* and other Coleoptera species indicates that TyunCSP1 clusters with DponCSP3, AglaCSP5, and TcasCSP4, while TyunCSP2 clusters with DvalCSP2, ItypCSP1, and DponCSP1. Additionally, other CSPs from *T. yunnanensis* also form distinct branches with CSPs from the aforementioned species (Appendix A). The phylogenetic tree shows that the highly conserved TyunOR35 clusters with Orcos from the *Tribolium castaneum* and *D. ponderosae* in the evolutionary tree. TyunOR35 clusters with Orco of *D. ponderosae* (Appendix A). Phylogenetic analysis of *T. yunnanensis* GRs shows that TyunGR24 clusters with members of the Sugar GR subfamily, including Agla64bSt, DponGR5NTE, Tcas64fSt, and AplaGR9JOI. TyunGR17 and TyunGR23 group with CO2 GRs. TyunGR13 clusters with DponGR47, which is the fructose receptor of the *D. ponderosae* (Appendix A). Phylogenetic analysis of 15 IRs shows that two candidate TyunIRs (TyunIR8, TyunIR34) are located in the IR75 subfamily branch. Moreover, 11 TyunIRs are in the iGluRs branch, with TyunIR7 and TyunIR17 being paralogous. TyunIR14 clusters with DponIR25a and belongs to the IR25 subfamily, while TyunIR11 clusters with DponIR21 and belongs to the IR21 subfamily (Appendix A).

To ensure the reliability of the phylogenetic tree, six short SNMP sequences were removed, and the two identified TyunSNMPs were subjected to tree analysis with SNMPs from other Coleoptera species. The results show that TyunSNMP8 clusters with DponSNMP1b and TyunSNMP4 clusters with DponSNMP1a, both belonging to the SNMP1 subfamily (Appendix A).

### 3.3. Compound Concentration Screening

To further clarify whether the chemosensory-related genes obtained from the identification are regulated by odor substances to participate in host localization, we conducted olfactory fumigation experiments [35]. In the compound concentration screening experiment using the odorants of *A. ferdinandi-coburgii* (non-host plant volatile: 4′-ethylacetophenone) and *P. yunnanensis* (host plant volatile: α-pinene), *T. yunnanensis* without any odor treatment served as the control. After 24 h of odor treatment in a 6 °C refrigerator, the mortality of *T. yunnanensis* was observed under a dissecting microscope. The observations revealed that the control group of beetles, without any odor treatment, had a 0% mortality rate after 24 h at 6 °C. At a concentration of 200 ppm, the mortality rate of beetles treated with α-pinene was approximately 43%, while those treated with 4′-ethylacetophenone had a mortality rate of about 53% (Figure 3). Through the concentration screening experiment, a concentration of 200 ppm, which resulted in a mortality rate close to 50%, was ultimately selected as the treatment concentration for subsequent experiments.

### 3.4. Expression of Chemosensory-Related Genes Was Affected by Odor Substances

Based on the results of chemosensory-related gene identification (high expression of OBP with antenna specificity) and the key CYPs (CYP4 and CYP6 families, tissue-specific high expression genes) obtained from the preliminary research of the project [32], we selected *TyunOBP6*, *TyunCYP4G2*, and *TyunCYP6DF1* for subsequent experiments. At 8 and 24 h of treatment, the expression levels of *TyunOBP6* in the antennae of male beetles treated with α-pinene were significantly higher than those treated with 4′-ethylacetophenone (*p* < 0.05) (Figure 4a,b). Under 8 h treatment conditions, *TyunCYP4G2* expression in the residual bodies of females treated with α-pinene was significantly higher than in those treated with 4′-ethylacetophenone, while in males, it was significantly higher with 4′-ethylacetophenone treatment than with α-pinene treatment (*p* < 0.05) (Figure 4c,d). *TyunCYP6DF1* expression in the residual bodies of females treated with 4′-ethylacetophenone was significantly lower than in those treated with α-pinene (*p* < 0.05) (Figure 4e,f). Under 24 h treatment conditions, the expression level of *TyunOBP6* in the antennae of female beetles treated with 4′-ethylacetophenone was significantly higher than in those treated with α-pinene (Figure 4a,b). The expression levels of *TyunCYP4G2* in the residual bodies of both male and female beetles treated with 4′-ethylacetophenone were significantly higher than in those treated with α-pinene (*p* < 0.05) (Figure 4c,d). *TyunCYP6DF1* expression in the residual bodies of both male and female beetles treated with 4′-ethylacetophenone was significantly lower than in those treated with α-pinene (*p* < 0.05) (Figure 4e,f). The above results indicate that host and non-host compounds regulate gene (*TyunOBP6*, *TyunCYP4G2*, and *TyunCYP6DF1*) expression of *T. yunnanensis* to different degrees.

### 3.5. Analysis of Binding Information Between Compounds and Proteins

The homology modeling consistency of TyunOBP6, TyunCYP4G2, and TyunCYP6DF1 of *T. yunnanensis* was more than 30%. The accuracy of model conformation was further evaluated using a ramachandran map. The proportion of amino acid residues falling within the allowed regions accounted for more than 90% of the entire protein, ranging from 90.43% to 97.98% (Appendix A). Regarding α-pinene, its free binding energy with the TyunOBP6 receptor was −4.03 kcal/mol (Figure 5a,b), its binding energy with the TyunCYP4G2 receptor was −5.46 kcal/mol (Figure 5c,d), and its binding energy with the TyunCYP6DF1 receptor was −5.18 kcal/mol (Figure 5e,f). The docking results with 4′-ethylacetophenone show that the free binding energy with the TyunOBP6 receptor was −3.83 kcal/mol (Figure 5g,h), the binding energy with the TyunCYP4G2 receptor was −5.45 kcal/mol (Figure 5i,j), and with the TyunCYP6DF1 receptor, it was −5.92 kcal/mol (Figure 5k,l). Both host and non-host compounds bind to the same “pocket” on the chemosensory-related proteins. TyunOBP6 and TyunCYP4G2 share the same amino acid binding sites (lysine 4, arginine 502) with host and non-host compounds, indicating that the non-host compounds are in direct competition or mutual exclusivity with the host compounds for the binding sites of OBP and CYP, affecting the insect host (Figure 5).

During molecular dynamics simulations, protein stability was monitored by calculating the RMSD. The RMSD values of each complex are shown in Figure 6a,c. The RMSF values, derived from atomic positional fluctuations in the trajectory, were used to evaluate complex flexibility, as depicted in Figure 6b,d. Integrated analysis of the RMSD and RMSF results indicates that TyunOBP6 exhibits higher stability with 4′-ethylacetophenone, whereas TyunCYP4G2 shows stronger binding stability with α-pinene. These conclusions are further supported by the Rg, SASA, and relative free energy distribution data (Appendix A).

## 4. Discussion

Insect chemosensory-related genes have been continuously refined through years of research, with an increasing number of genes being identified and excavated. As research progresses, it has become clear that the number and function of chemosensory genes in different insects also show significant differences. These differences are believed to be related to specific lifestyles and environmental adaptations, and may ultimately lead to the formation of new species [43,44]. Studies also predict that polyphagous insect species have more chemosensory genes than host-specific oligophagous insects. Among Coleoptera insects, there is also a huge difference in the number of chemosensory-related genes. In *Tribolium castaneum*, 610 chemosensory-related genes were identified, including 50 *OBPs*, 20 *CSPs*, 341 *ORs*, 15 *SNMPs*, 143 *CYPs*, and 41 *GSTs* [45,46,47,48]. In *D*. *ponderosae*, a considerable number of chemosensory genes were also identified, including 86 *ORs*, 60 *GRs*, 57 *IRs*, 4 *SNMPs*, 36 *OBPs*, 11 *CSPs*, and 47 *CYPs* [49]. In the study of *T. yunnanensis*, Liu Naiyong and others identified 45 *OBPs*, 12 *CSPs*, 20 *ORs*, 8 *GRs*, 3 *IRs*, and 3 *SNMPs* from the beetles at different developmental stages [50]. Zhu Jiaying and others identified 11 *OBPs*, 8 *CSPs*, 18 *ORs*, and 8 *GRs* from the transcriptome of *T. yunnanensis* [25]. Our study has enriched the work on the chemosensory related genes of *T. yunnanensis*. By integrating transcriptomic data across different developmental stages and genomic annotations, this study further enriches the chemosensory gene repertoire of *T. yunnanensis*. Combining the explorations of previous articles with the analysis of our results, most of the chemosensory-associated proteins are not simply expressed only in chemosensory-related organs; in fact, this conclusion also confirms that chemosensory-associated proteins in insects, in addition to participating in the recognition of environmental chemical signals as a core function, also have corresponding functional values for reproduction and immunity [51,52].

Compared to previous studies, our work significantly expands the number and diversity of identified genes through more comprehensive sequencing strategies and functional annotations, laying a solid foundation for subsequent functional investigations. In a comparative transcriptome study of *T. yunnanensis*, *Tomicus brevipilosus,* and *Tomicus minor*, Ting-Ting Lu et al. clarified that members of *carboxylesterases*, *IRs*, *SNMPs* and *iGluR* are highly conserved in terms of the number of genes and sequence identity among the three Tomicus species. However, the expression of genes related to reproduction was different [26]. There are differences in the chemosensory-related genes identified among different species of Coleoptera, which may be due to the influence of environmental or other factors during the evolutionary process. And the more similar the relatives or habits, the higher the consistency of chemosensory-related genes in the species. *T. yunnanensis*, *D. ponderosae,* and *Dendroctonus armandi* are all bark beetles, and their genes have high homology, with the number of genes identified also varying [53,54]. Over a long period of evolution, bark beetles have developed their unique survival characteristics, and their host plants are different. The different stimuli from plant volatiles also lead to various evolutionary trends in their chemosensory and ODE genes.

Our research focuses on *TyunOBP6*, a gene that is highly expressed in the antennae of *T. yunnanensis*, suggesting that it is an antenna-specific OBP and may be involved in host location. In the phylogenetic analysis of TyunOBP6, it was found to have a high degree of homology with DponOBP5 and clusters with DponOBP6a and DponOBP6b in the ABPII branch. In previous reports, DponOBP6a/b alternately encodes the first exon of the signal peptide, which is a hydrophobic motif that allows the protein to be expelled from the cell [53]. DponOBP6 being an antenna-specific protein provides strong support for the speculation that TyunOBP6 is antenna-specific [55]. When exposed to chemicals, the regulated expression of OBP genes in the antennae of *T. yunnanensis* is consistent with the regulated expression of AgrnOBPs in the antennae of the cotton boll weevil, indicating that these systemic expression changes of OBPs may be related to improving the detection of signaling chemical stimuli [56]. However, both α-pinene and 4′-ethylacetophenone can induce the expression of *TyunOBP6*, indicating that compound induction treatment methods help identify functional OBPs associated with odor pairing, and the specific definition of their function still requires further determination using a variety of methods including molecular docking, fluorescence competitive binding, and others [57,58].

As a response to attacks by bark beetles, host plants greatly increase the concentration of compounds they release after being attacked, leading to a higher mortality rate for bark beetles [59]. As observed in compound concentration selection experiments, the type and concentration of compounds seem to affect the survival rate of bark beetles. At high concentrations, all compounds lead to a higher mortality rate in *T. yunnanensis*, and there is little difference in the lethality of host and non-host compound concentrations in the beetles. Perhaps non-host volatiles do not directly cause the beetles to die but interfere with their ability to find hosts, resulting in failure in obtaining food resources and thus they cannot continue to survive [60,61]. Under natural conditions, bark beetles absorb compounds through the respiratory system, cuticle, and digestive tract. However, since the experiment used odor fumigation rather than feeding strategies for *T. yunnanensis*, the induction of detoxification genes observed in the remnants of bark beetles is mainly a response to the inhalation of these compounds. López and others found that unfed *Dendroctonus valens* exposed to the odors of plant secondary metabolites showed an increase in the number of secretory vesicles, lysosomes, and mitochondria, changes in the inner and outer membranes of the mitochondria, a large presence of a smooth endoplasmic reticulum, and other ultrastructural changes, indicating that the midgut of this species is in a detoxification state [62,63,64]. In our study, *TyunCYP4G2* is specifically highly expressed in the remnants of *T. yunnanensis*, which may be related to the need for *T. yunnanensis* to metabolize toxic substances in the gut and other parts. Other studies have found that the main biochemical function of *CYP4* family genes in bark beetles may not only be the degradation or transformation of exogenous monoterpenes [62,65]. Evidence from other insect systems suggests that *CYP4* family genes are involved in the hydroxylation of fatty acids, as well as the biosynthesis and metabolism of pheromones [65]. The CYP6 family is unique to insects, and its biochemical function is essentially related to the metabolism of plant chemicals in phytophagous insects. However, little is known about its function in bark beetles [66,67]. In many studies, genes of the CYP6 family have been shown to metabolize exogenous substances and plant natural compounds [68]. In our study, *TyunCYP6DF1* is highly expressed in the remnants, and it is very likely to be involved in the detoxification of plant secondary metabolites. However, in compound treatment experiments, its expression is significantly downregulated, suggesting that it may not be involved in the detoxification function of α-pinene and 4′-ethylacetophenone. This conclusion still needs to be explored, as the degradation or action of a single compound may itself follow multiple metabolic elimination pathways. In addition, the inhibition of enzyme expression does not constitute clear evidence of enzyme non-participation, and there may be an alternative pathway that is conducive to preventing the accumulation of potentially toxic intermediates [69]. Molecular docking and molecular dynamics simulation analyses further validated our conjecture that there is a potential influence of non-host compounds on host compounds in binding to proteins. TyunOBP6 and TyunCYP4G2 have the same amino acid binding sites for host and non-host compounds, indicating that non-host compounds can compete with host compounds for the binding sites of olfactory proteins. However, there are still some gaps in our research, and we will follow up with experimental verification of the function of key genes and a deeper analysis by simulating the state of compounds in natural conditions and conducting experiments on insect behavior.

## 5. Conclusions

This study identified 137 chemosensory-related genes of *T. yunnanensis*, including their sequence characteristics and tissue expression profiles. Building on previous research that identified key compounds required for *T. yunnanensis* to locate its host, *P. yunnanensis*, and the volatiles of the non-host plant (*A. ferdinandi-coburgii*), which significantly affect *T. yunnanensis*, we used molecular docking to determine the binding sites and modes of host and non-host compounds with *TyunOBPs* and *TyunCYPs*, clarifying that there is competition for binding sites among the compounds which affects their host localization.

## Figures and Tables

**Figure 1 insects-16-00553-f001:**
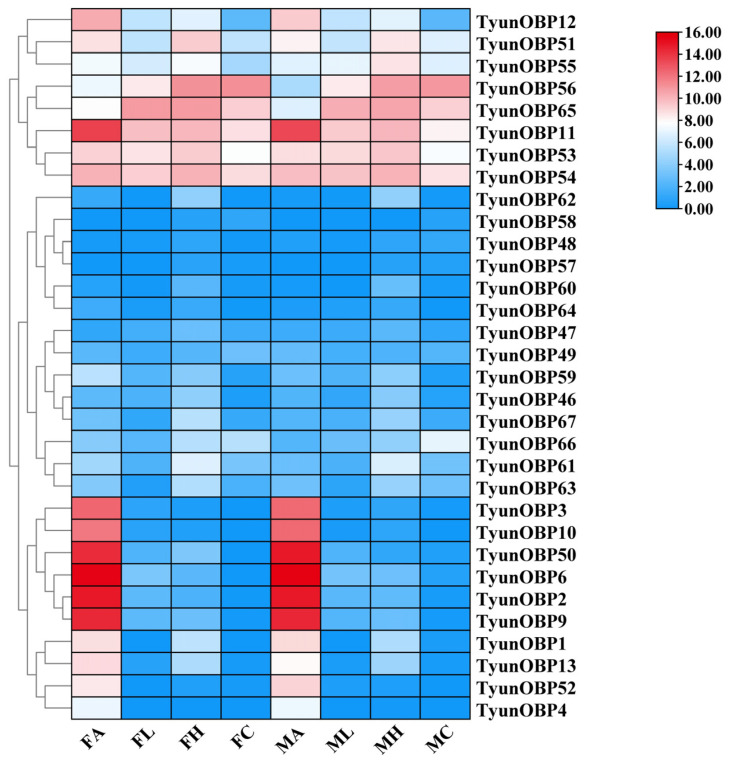
Expression profiles of *TyunOBP* genes in *T. yunnanensis*. FA: female antennae; MA: male antennae; FH: female head; MH: male head; FL: female leg; ML: male leg; FC: female carcass; MC: male carcass.

**Figure 2 insects-16-00553-f002:**
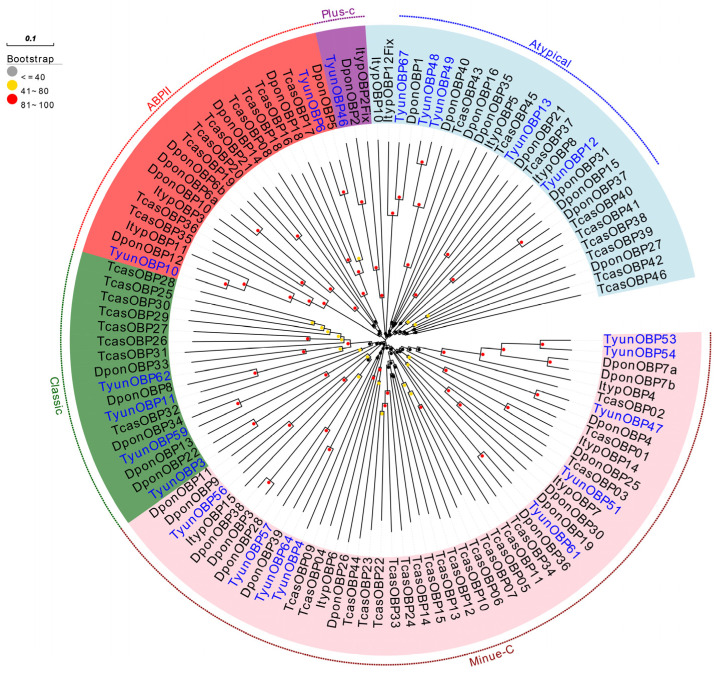
Neighbor-joining tree of candidate TyunOBPs. Bootstrap values after 1000 replications. Dpon, *Dendroctonus ponderosae*; Ityp, *Ips typographus*; Tcas, *Tribolium castaneum*; Tyun, *Tomicus yunnanensis*.

**Figure 3 insects-16-00553-f003:**
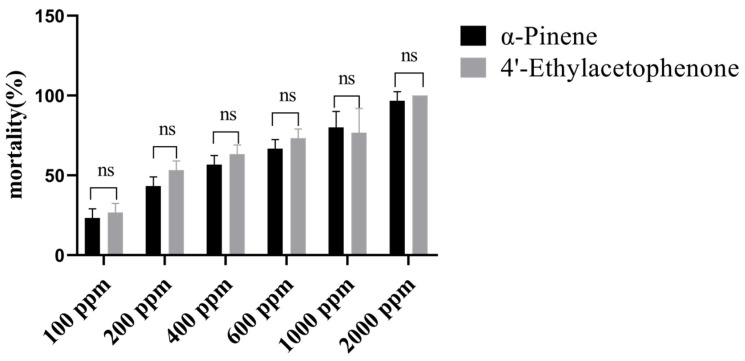
Concentration screening test results. The ns represents no significant difference. *X*-axis indicates the concentration of the added compound and *Y*-axis indicates mortality rate of *T. yunnanensis*.

**Figure 4 insects-16-00553-f004:**
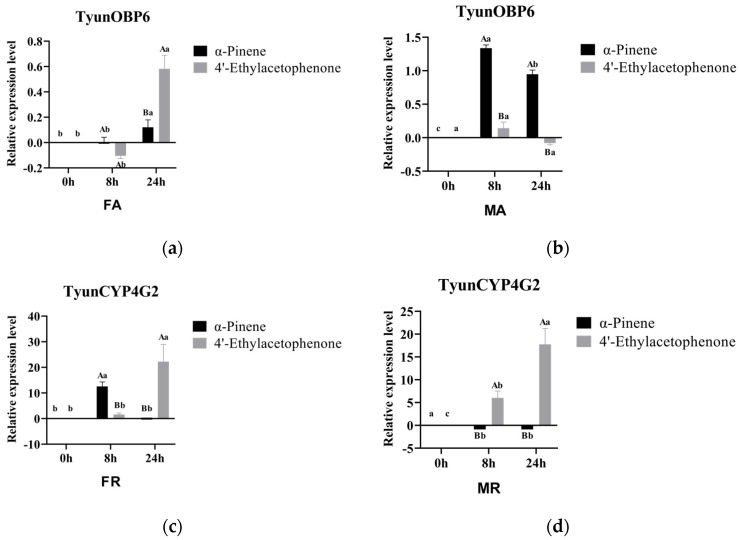
Analysis of *TyunOBP* and *TyunCYP* gene expression after compound treatment. By one-way ANOVA, different lowercase letters indicate that the difference between 0 h, 8 h, and 24 h in the same stimulus was statistically significant (*p* < 0.05), and different uppercase letters indicate that the difference between different stimuli was statistically significant (*p* < 0.05). (**a**) *TyunOBP6* gene expression of female antennae; (**b**) *TyunOBP6* gene expression of male antennae; (**c**) *TyunCYP4G2* gene expression of female remain; (**d**) *TyunCYP4G2* gene expression of male remain; (**e**) *TyunCYP6DF1* gene expression of female remain; (**f**) *TyunCYP6DF1* gene expression of male remain.

**Figure 5 insects-16-00553-f005:**
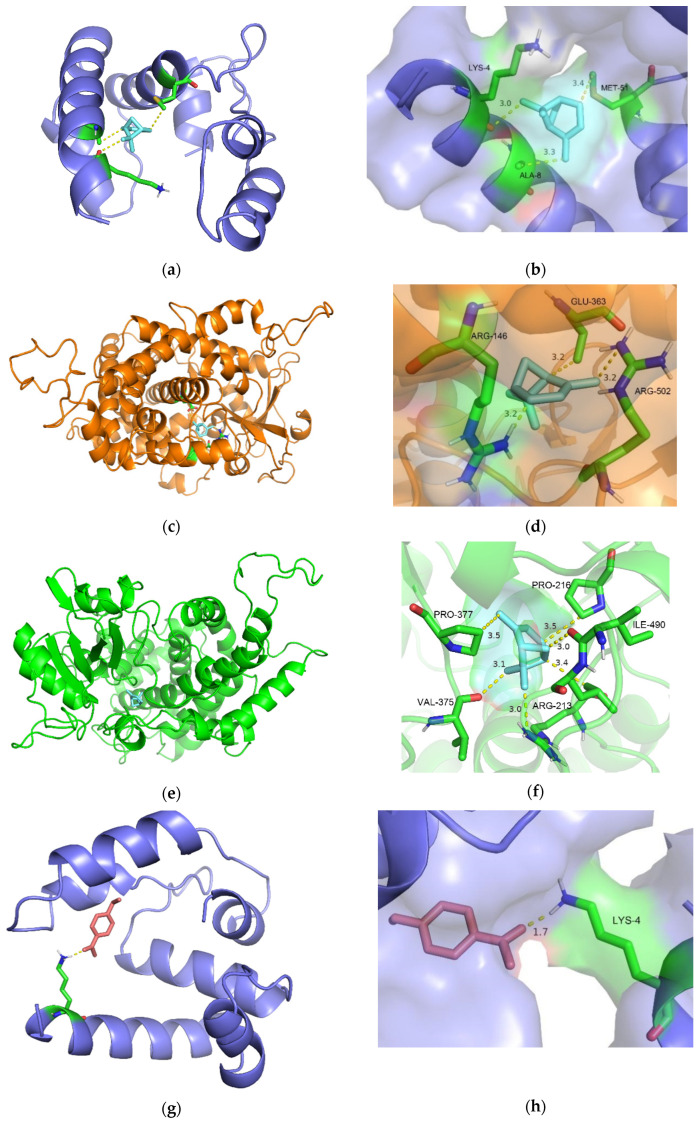
Docking pattern of protein and α-pinene or 4′-ethylacetophenone of T. yunnanensis. (**a**,**b**) the molecular docking pattern of TyunOBP6 with α-pinene; (**c**,**d**) the molecular docking pattern of TyunCYP4G2 with α-pinene; (**e**,**f**) the molecular docking pattern of TyunCYP6DF1 with α-pinene; (**g**,**h**) the molecular docking pattern of TyunOBP6 with 4′-ethylacetophenone; (**i**,**j**) the molecular docking pattern of TyunCYP4G2 with 4′-ethylacetophenone; (**k**,**l**) the molecular docking pattern of TyunCYP6DF1 with 4′-ethylacetophenone.

**Figure 6 insects-16-00553-f006:**
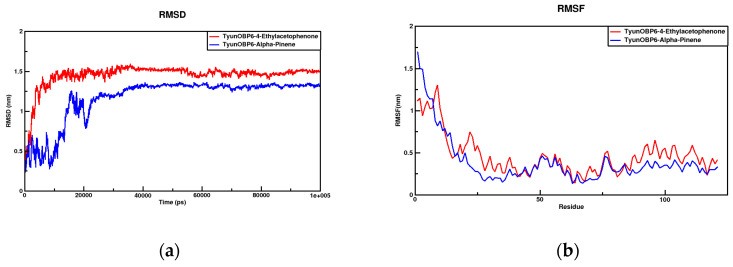
Molecular dynamics simulations showing the RMSD and RMSF results of the complex. (**a**) The red color represents the RMSD results of the complex formed between TyunOBP6 and 4′-ethylacetophenone, while the blue color represents the RMSD results of the complex formed between TyunOBP6 and α-pinene; (**b**) the red color represents the RMSF results of the complex formed between TyunOBP6 and 4′-ethylacetophenone, while the blue color represents the RMSF results of the complex formed between TyunOBP6 and α-pinene; (**c**) the red color represents the RMSD results of the complex formed between TyunCYP4G2 and 4′-ethylacetophenone, while the blue color represents the RMSD results of the complex formed between TyunCYP4G2 and α-pinene; (**d**) the red color represents the RMSF results of the complex formed between TyunCYP4G2 and 4′-ethylacetophenone, while the blue color represents the RMSF results of the complex formed between TyunCYP4G2 and α-pinene.

## Data Availability

All of the data are available in the NCBI database under the project ID PRJNA1253926.

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
