# Peer review of "Mechanisms of Impact of Alnus ferdinandi-coburgii Odor Substances on Host Location of Tomicus yunnanensis"

_insects, 2025, doi:10.3390/insects16060553_

Round 1

Reviewer 1 Report

Comments and Suggestions for Authors

This study employs an integrated approach combining transcriptomic analysis, gene expression profiling, and molecular docking to identify chemosensory-related genes in Tomicus yunnanensis, a major pest of Pinus yunnanensis. The research aims to elucidate how non-host plant volatiles from Alnus ferdinandi-coburgii interfere with the host location behavior of T. yunnanensis. However, some methodological and interpretive aspects could be strengthened.

  1. While the docking results suggest competition for binding sites, the conclusions would be more robust with in vitro or in vivo validation (e.g., through competitive binding assays). The relatively low free binding energy values (e.g., -3.83 kcal/mol) suggest that the interactions between OBPs and ligands may be weak. Also, molecular docking alone is insufficient for definitive conclusions. The molecular dynamics simulations are recommended to validate the docking results.
  2. The molecular docking results seem overinterpreted in this manuscript. Lines 319-322, the host and non-host compounds shared the same binding sites. Please clarify why the non-host compounds can compete with host compounds for the binding site. What are the biological functions of this competence mechanism? Again, are the observed binding energies sufficient to support the proposed competitive mechanism under physiological conditions?  
  3. The use of mortality rates (Figure 3) as an indicator of repellency is problematic, as mortality more accurately reflects chemical toxicity rather than behavioral repellence. Commonly, even attractive compounds can be lethal at high concentrations. Therefore, mortality data alone cannot conclusively demonstrate repellent properties. Please clarify this.
  4. Line 282-283, please explain why these three genes were selected for this study.

Author Response

Dear Reviewer:

We sincerely appreciate the reviewers’ valuable comments and suggestions. We have carefully addressed all the concerns and revised the manuscript accordingly. Below are our point-by-point responses, with the changes highlighted in the revised manuscript.

  1. While the docking results suggest competition for binding sites, the conclusions would be more robust with in vitro or in vivo validation (e.g., through competitive binding assays). The relatively low free binding energy values (e.g., -3.83 kcal/mol) suggest that the interactions between OBPs and ligands may be weak. Also, molecular docking alone is insufficient for definitive conclusions. The molecular dynamics simulations are recommended to validate the docking results.

Return

We have supplemented molecular dynamics simulations:During molecular dynamics simulations, protein stability was monitored by calculating RMSD. The RMSD values of complex are shown in Figures 6a and 6c. RMSF values, derived from atomic positional fluctuations in the trajectory, were used to evaluate complex flexibility, as depicted in Figures 6b and 6d. Integrated analysis of RMSD and RMSF results indicated that TyunOBP6 exhibits higher stability with 4'-ethylacetophenone, whereas TyunCYP4G2 shows stronger binding stability with α-pinene. These conclusions are further supported by Rg, SASA, and relative free energy distribution data.

  1. The molecular docking results seem overinterpreted in this manuscript. Lines 319-322, the host and non-host compounds shared the same binding sites. Please clarify why the non-host compounds can compete with host compounds for the binding site. What are the biological functions of this competence mechanism? Again, are the observed binding energies sufficient to support the proposed competitive mechanism under physiological conditions?

Return:

The binding sites of both (molecules/ligands) are located in the same hydrophobic pocket and share key residues, suggesting potential direct competition or mutual exclusion. Direct competition implies that the two odorant molecules may compete for binding to the same site, where the binding of one inhibits the other. Mutual exclusion means that if a non-host odorant occupies the protein's binding site, it may prevent the host odorant from being recognized by the insect, thereby disrupting host location and achieving pest resistance. Molecular docking currently supports this conclusion, and further protein experiments are planned to validate this process. For scientific rigor, this section has been revised accordingly.

  1. The use of mortality rates (Figure 3) as an indicator of repellency is problematic, as mortality more accurately reflects chemical toxicity rather than behavioral repellence. Commonly, even attractive compounds can be lethal at high concentrations. Therefore, mortality data alone cannot conclusively demonstrate repellent properties. Please clarify this.

Return:

We acknowledge that the chemical toxicity you mentioned cannot represent repellent properties. For the objectives of this study, repellency is not the research focus. Instead, we aim to elucidate the molecular mechanisms of insect resistance in mixed forests by exploring the association between compounds and chemosensory-related proteins. Our goal is not repellency-oriented but rather to interfere with Tomicus yunnanensis's recognition of host plants to achieve pest resistance. Here, the selection of chemically toxic doses for treatment is solely to rely on high concentrations to determine whether the compounds significantly affect protein expression.

  1. Line 282-283, please explain why these three genes were selected for this study.

Return:

OBP6 is an antenna-specific protein with the highest expression level. Two genes from the CYP4and CYP6 families were described in our previous study: members of the CYP4 family are involved in odorant degradation, with most participating in detoxification and synergistic reactions in pheromone biosynthesis. TyunCYP4G2 is specifically highly expressed in residues, possibly related to the metabolism of toxic substances in Tomicus yunnanensis. Genes from the CYP6subfamily have been shown to metabolize xenobiotics and plant-derived natural compounds. Their expression levels and gene family characteristics were the primary reasons for their selection.

Added content in the Methods section: In the selection of key genes for subsequent research, the OBP genes were prioritized based on their antennal-specific high expression patterns. The CYP genes were selected based on both their high tissue expression levels and their classification into the CYP4 and CYP6 subfamilies, which are closely associated with insect-mediated degradation of plant secondary metabolites.

Added content in the Results section: Based on the results of chemosensory-related gene identification (High expression of OBP with antenna specificity) and the key CYPs (CYP4 and CYP6 families, tissue-specific high expression genes) obtained from the preliminary research of the project, we have selected TyunOBP6, TyunCYP4G2, and TyunCYP6DF1 for subsequent experiments.

Reviewer 2 Report

Comments and Suggestions for Authors

The manuscript requires careful editing to improve its form and eliminate typographical and formatting issues. For instance, on line 34, the first word of the introduction is incorrectly written as "Tthe". Another example can be found on line 54, where a space is missing before "OBPs". A thorough proofreading of the manuscript is necessary to correct these kinds of errors.

On line 54, OBPs are described as odorant transporters. However, this role should be presented more cautiously. To date, no study has demonstrated a direct interaction between OBPs and olfactory receptors (ORs). Furthermore, OBPs may also serve a protective role by binding to odorant molecules and shielding them from premature degradation by odorant-degrading enzymes (ODEs). This more nuanced perspective should be reflected in the text. For a balanced view of OBP function, the following publications should be cited: https://doi.org/10.7554/eLife.20242.001, doi.org/10.3389/finsc.2023.1274197

Line 67: The list of CSPs (chemosensory proteins) should be revised to include UDP-glycosyltransferases, as recent findings support their role in insect olfaction. Their involvement in olfactory processes has been demonstrated, and the following publication provides strong evidence: doi: 10.3390/genes11030237. This addition would improve the completeness and accuracy of the classification.

In addition, references should be added to support the olfactory functions of various ODE families. For example, the following review provides a useful synthesis of current knowledge on GSTs in insect olfaction and would be an appropriate citation: DOI: 10.3390/biom13020322.

Line 194: The study would have benefited from an ortholog search—using Drosophila as a model, for instance—to help identify the specific CSPs and clarify their enzymatic nature. This would add functional insight and contextual depth to the analysis.

Line 282: The selection criteria for the three enzymes analyzed are not clearly explained. Detoxification of xenobiotic compounds typically involves phase I enzymes followed by phase II enzymes, or sometimes exclusively phase II enzymes. Why were phase II enzymes not included in the analysis? Why were these two specific CYPs and this particular OBP selected? Please clarify the selection process and better explain how reference 33 supports this choice.

Additionally, in Figure 5, referring to the structural model as a “diagram” does not seem appropriate; a more accurate term should be used to describe the structural modeling.

Regarding the Discussion section, the proposed functions should once again be presented with greater caution and nuance. The olfactory role in particular should be suggested without disregarding other potential functions.

Still within the Discussion, the interpretation of the results shown in Figure 4 should be tempered by two key considerations: (i) A single compound may follow multiple metabolic elimination pathways. (ii) The repression of an enzyme’s expression does not constitute definitive evidence that the enzyme is not involved—it could also indicate that an alternative pathway is favored to prevent the accumulation of a potentially toxic intermediate.

Author Response

Dear Reviewer:

We sincerely appreciate the reviewers’ valuable comments and suggestions. We have carefully addressed all the concerns and revised the manuscript accordingly. Below are our point-by-point responses, with the changes highlighted in the revised manuscript.

1.The manuscript requires careful editing to improve its form and eliminate typographical and formatting issues. For instance, on line 34, the first word of the introduction is incorrectly written as "Tthe". Another example can be found on line 54, where a space is missing before "OBPs". A thorough proofreading of the manuscript is necessary to correct these kinds of errors.

Return: All modification checks have been completed

2.On line 54, OBPs are described as odorant transporters. However, this role should be presented more cautiously. To date, no study has demonstrated a direct interaction between OBPs and olfactory receptors (ORs). Furthermore, OBPs may also serve a protective role by binding to odorant molecules and shielding them from premature degradation by odorant-degrading enzymes (ODEs). This more nuanced perspective should be reflected in the text. For a balanced view of OBP function, the following publications should be cited: https://doi.org/10.7554/eLife.20242.001, doi.org/10.3389/finsc.2023.1274197

Return:

This part of the revision has been completed: Typically, external odorants or volatile compounds pass through cuticular pores into the sensillum lymph, where they are bound and solubilized by OBPs. These OBP-odorant complexes are then transported through the sensillum lymph to potential receptors, enabling signal transduction.

  1. Line 67: The list of CSPs (chemosensory proteins) should be revised to include UDP-glycosyltransferases, as recent findings support their role in insect olfaction. Their involvement in olfactory processes has been demonstrated, and the following publication provides strong evidence: doi: 10.3390/genes11030237. This addition would improve the completeness and accuracy of the classification.

In addition, references should be added to support the olfactory functions of various ODE families. For example, the following review provides a useful synthesis of current knowledge on GSTs in insect olfaction and would be an appropriate citation: DOI: 10.3390/biom13020322.

Return:

We acknowledge that UDP-glycosyltransferases (UGTs) play an important role in insect chemosensation. However, the identification of this protein in coleopteran species remains limited and has not been included in our current study, which is why it was not discussed here. Nevertheless, we agree with your perspective and have added relevant descriptions regarding UGTs in the Introduction section. Additionally, we have supplemented the content on GSTs (glutathione S-transferases) based on the literature you provided and included further descriptions of CYPs (cytochrome P450s) and CCEs (carboxyl/cholinesterases).

4.Line 194: The study would have benefited from an ortholog search—using Drosophila as a model, for instance—to help identify the specific CSPs and clarify their enzymatic nature. This would add functional insight and contextual depth to the analysis.

Return:

Thank you for your suggestions. We have supplemented this section accordingly. In the gene identification of Tomicus yunnanensis, our homology search selected Tribolium castaneum (the model coleopteran insect) and closely related species of T. yunnanensis.

5.Line 282: The selection criteria for the three enzymes analyzed are not clearly explained. Detoxification of xenobiotic compounds typically involves phase I enzymes followed by phase II enzymes, or sometimes exclusively phase II enzymes. Why were phase II enzymes not included in the analysis? Why were these two specific CYPs and this particular OBP selected? Please clarify the selection process and better explain how reference 33 supports this choice.

Return:

We initially focused on phase I enzymes for preliminary investigation, with plans to subsequently explore phase II enzymes. However, the current research still faces significant technical challenges, as complete identification of phase II enzymes in homologous species remains relatively limited. Regarding our selection of CYPs and OBPs: OBP6 was chosen as it is an antenna-specific protein with the highest expression level. As for the two genes from CYP4 and CYP6 families, our previous publication described that CYP4 family members are involved in odorant degradation, with most participating in detoxification and synergistic reactions of pheromone biosynthesis. TyunCYP4G2 shows specific high expression in residues, potentially associated with toxic compound metabolism in Tomicus yunnanensis. Genes from the CYP6 subfamily have been demonstrated to metabolize xenobiotics and plant-derived natural compounds. Their expression levels and gene family characteristics were the primary criteria for their selection.

6.Additionally, in Figure 5, referring to the structural model as a “diagram” does not seem appropriate; a more accurate term should be used to describe the structural modeling.

Return:

This part of the revision has been completed: Docking pattern of protein and α-pinene or 4'-ethylacetophenone of T. yunnanensis. a-b:The molecular docking pattern of TyunOBP6 with α-pinene; c-d: The molecular docking pattern of TyunCYP4G2 with α-pinene; e-f: The molecular docking pattern of TyunCYP6DF1 with α-pinene; g-h: The molecular docking pattern of TyunOBP6 with 4'-ethylacetophenone; i-j: The molecular docking pattern of TyunCYP4G2 with 4'-ethylacetophenone; k-l: The molecular docking pattern of TyunCYP6DF1 with 4'-ethylacetophenone;

7.Regarding the Discussion section, the proposed functions should once again be presented with greater caution and nuance. The olfactory role in particular should be suggested without disregarding other potential functions.

Return:

Thank you for your valuable comments, we have tweaked parts of the discussion.

8.Still within the Discussion, the interpretation of the results shown in Figure 4 should be tempered by two key considerations: (i) A single compound may follow multiple metabolic elimination pathways. (ii) The repression of an enzyme’s expression does not constitute definitive evidence that the enzyme is not involved—it could also indicate that an alternative pathway is favored to prevent the accumulation of a potentially toxic intermediate.

Return:

Thank you for your valuable comments, which have been added to the discussion section.

Reviewer 3 Report

Comments and Suggestions for Authors

Dear Authors, Please find my comments in the attached file. 

Author Response

Dear Reviewer:

We sincerely appreciate the reviewers’ valuable comments and suggestions. We have carefully addressed all the concerns and revised the manuscript accordingly. Below are our point-by-point responses, with the changes highlighted in the revised manuscript.

Key concerns:

1.The candidate genes from the key subfamilies are missing (ORs, IRs and SNMP phylogenies). Discuss the limitations in the manuscript.

Return: Thank you for your valuable comments, this section has been revised and clarified.

2.The phylogenetic analysis method used was not optimal. ML trees based on evolutionary    models and with appropriate outgroups are highly recommended. However, the OBPs and CYPs seems to have good coverage and since the work is mainly focused on that, the

manuscript still holds its relevance.

Return: Thank you for your valuable advice.

Under identical input data, the Neighbor-Joining (NJ) method demonstrates high reproducibility in results. It directly constructs branch lengths from distance matrices, facilitating rapid interpretation. In this study, NJ was selected specifically to quickly obtain reasonable evolutionary hypotheses rather than pursuing statistically optimal solutions. For subsequent research involving larger species datasets, we plan to employ Maximum Likelihood (ML) methods for more precise analysis.

3.The methods need more clarity and detail. The preliminary research works needs to be cited at the beginning of the work, in order to avoid confusion.

Return: Thank you for your valuable advice, this part of the revision has been completed.

4.How you defined LC50 for bioassays? Were there enough data (or pilot studies) withAlnus ferdinandi-coburgii odor substances? Please provide details. Mortality rates are usually

represented by Kaplan-Meier survival plots.

Return:

In this study, the LC50 (median lethal concentration) was defined as the chemical concentration that caused 50% mortality in test insects under a fixed exposure duration (24 hours). The determination method followed [the gold standard approach has been to estimate a mass-specific LD50 (lethal dose 50): i.e. the dose (per unit body mass) that kills 50% of the population. DOI: 10.1098/rspb.2023.2811]. Mortality assessment in this experiment was based on endpoint observation at the fixed exposure time (24 hours), rather than continuous monitoring of time-dependent mortality dynamics. Under this design:

  • Kaplan-Meier survival plots are typically used to analyze time-dependent survival rate changes , while Single time-point mortality data can be directly calculated as proportions.

5.Minor concerns: (might repeat the key concerns)

L34: TT

Return: Thank you for your valuable advice, this part of the revision has been completed.

L58: What about other types of OBPs? Any tetramer OBPs?

Return: Thank you for your valuable advice, this part of the revision has been completed.

There is a group of OBPs that has been termed “atypical OBPs” characterized by 10 or more conserved cysteines, a long C-terminus, a conserved proline residue, and four or more disulfide bridges, which is recorded in several mosquito and locust species, suggesting this group of genes may be recently evolved in these species. 

L107: Any RNA-seq details? or citations?

Return: Thank you for your valuable advice, this part of the revision has been completed.

L125: Any details on specific species included in the annotation dataset? It will be good to    provide the species information to ensure that, the authors have included most of the well- annotated bark beetle ORs. This comment specifically refers to the low OR repertoire (36)    reported in T. yunnanensis!

Return: Thank you for your suggestions. We have supplemented this section accordingly. In the gene identification of Tomicus yunnanensis, our homology search selected Tribolium castaneum (the model coleopteran insect) and closely related species of T. yunnanensis.

L146: Why neighbor-joining (NJ) method was applied for all chemosensory related proteins? Authors might be knowing that ML trees based on well-supported amino acid substitution

models have already used to define and classify most Coleopteranchemosensory gene    families. Based on the supplementary Figures S1A-E, phylogenetic analysis of mainly ORs and IRs are insufficient and has methodological errors.

Return:

We acknowledge the established authority of Maximum Likelihood (ML) in phylogenetic reconstruction. However, the Neighbor-Joining (NJ) method remains widely employed in phylogenetic analyses of certain taxa and fully meets the requirements of our current study.

L151: How you calculated α-pinene and 4'-ethylacetophenone LC50s. Are the tested compounds listed in Reference 31? Was there any pilot experiment?

Return:

We appreciate your suggestions. In this study, we did not focus on the repellent properties of the compounds. Preliminary experiments confirmed that non-host compounds affect insect growth and development. Building on this, we aimed to demonstrate that non-host compounds interfere with the insect's ability to recognize host compounds. Here, we selected chemically toxic doses for treatment to determine whether the compounds significantly affect protein expression by relying on high concentrations. Mortality rates (LC50) were determined based on the actual counts of surviving and dead insects in the experiments.

L160: Why the study was conducted at 6°C? any references? Please cite.

Return:

In the absence of feed, in order to maintain the basic viability of the T. yunnanensis and to reduce their activity, it was explored and found that a natural survival of 24 hours at 6 degrees Celsius could be ensured.

L169: How was β-actin selected? any reference? Two endogenous controls are

recommended for expression profiling studies. Provide Tm details in supplementary info.

Return:

The candidate reference genes selected for this study were based on the article PMID: 34079474. Preliminary experiments revealed that only β-actin exhibited stable expression across different stimulation conditions, and was therefore chosen as our internal reference gene. This information has been supplemented in the manuscript.

L171: Method used for calculating relative gene expression levels? Or reference for Q-Gene?

Return: Thank you for your suggestions. We have supplemented this section accordingly:qPCR cycling parameters were: 94 °C for 4 min followed by 40 cycles at 94 °C for 20 s and 60 °C for 30 s. Relative gene expression level was calculated using the Q-GENE statistical analysis package (According to the formula: relative gene expression=)

L176: Why homology modeling was used over AlphaFold?

Return: Tool Selection Reference:Yandamuri RC, Gautam R, Darkoh C, Dareddy V, El-Bouhssini M, Clack BA. Cloning, Expression, Sequence Analysis and Homology Modeling of the Prolyl Endoprotease from Eurygaster integriceps Puton. Insects. 2014 Oct 22;5(4):762-82. doi: 10.3390/insects5040762.

Zhan E, Jiang J, Wang Y, Zhang K, Tang T, Chen Y, Jia Z, Wang Q, Zhao C. Shisa reduces the sensitivity of homomeric RDL channel to GABA in the two-spotted spider mite, Tetranychus urticae Koch. Pestic Biochem Physiol. 2023 May;192:105414. doi: 10.1016/j.pestbp.2023.105414

The choice of homology modeling in the study ensures higher accuracy of insect proteins.

L186: Missing details. Any protocol for docking and visualization? Please cite.

Return: Thank you for your suggestions. We have supplemented this section accordingly: Molecular docking was performed separately with the two compounds using AutoDock4.0. The docking pocket was designed based on a grid box set as the largest cube enclosing the protein, while AutoDock4.0 automatically identified the central binding pocket, and the top ranked binding mode was evaluated according to the docking score, and visually analyzed by PyMOL (version 1.9.0).

L244-256: The phylogenetic trees and the classification of ORs, IRs and GRs only highlights

the limitations of your study in terms of annotation and phylogenetic analysis. ML trees with outgroups are preferred for most chemosensory genes. Is TyunOR35 an ORCo? Without OR-  coreceptor (ORco), OR annotations make less sense. Same applies to IRs and IR-coreceptors.

It will be much better to provide sequence information of annotated genes as supplementary info.

Return:

Our analysis clearly identified TyunOR35 as belonging to the ORco family. For IR annotations, we have specifically labeled the iGluR subgrou. For example: The phylogenetic tree showed that the highly conserved TyunOR35, clusters with Orcos from the Tribolium castaneum and D. ponderosae in the evolutionary tree. TyunOR35 was orthologous to the Orco of the D. ponderosae (Supplementary Figure S1B). In addition, because of the research focus on OBP, work on proteins such as OR has been placed in a later program.

L282: Preliminary research of the project published as citation numbered 33 is an important information missed in the methods section (mainly for CYPs?). Similarly, what about the

citation number 23 (for OBPs)?

Return:

OBP6 is an antenna-specific protein with the highest expression level. Two genes from the CYP4and CYP6 families were described in our previous study: members of the CYP4 family are involved in odorant degradation, with most participating in detoxification and synergistic reactions in pheromone biosynthesis. TyunCYP4G2 is specifically highly expressed in residues, possibly related to the metabolism of toxic substances in Tomicus yunnanensis. Genes from the CYP6subfamily have been shown to metabolize xenobiotics and plant-derived natural compounds. Their expression levels and gene family characteristics were the primary reasons for their selection.

Added content in the Methods section: In the selection of key genes for subsequent research, the OBP genes were prioritized based on their antennal-specific high expression patterns. The CYP genes were selected based on both their high tissue expression levels and their classification into the CYP4 and CYP6 subfamilies, which are closely associated with insect-mediated degradation of plant secondary metabolites.

Added content in the Results section: Based on the results of chemosensory-related gene identification (High expression of OBP with antenna specificity) and the key CYPs (CYP4 and CYP6 families, tissue-specific high expression genes) obtained from the preliminary research of the project, we have selected TyunOBP6, TyunCYP4G2, and TyunCYP6DF1 for subsequent experiments.

L320: Such details like defining the “ binding pocket” are missing in methods section.

Return: Thank you for your suggestions. We have supplemented this section accordingly: Molecular docking was performed separately with the two compounds using AutoDock4.0. The docking pocket was designed based on a grid box set as the largest cube enclosing the protein, while AutoDock4.0 automatically identified the central binding pocket, and the top ranked binding mode was evaluated according to the docking score, and visually analyzed by PyMOL (version 1.9.0).

L355: Did you mean that numbers are conserved except for ORs? would that bean annotation related issue?

Return:

This section's introduction is solely based on previous studies of carboxylesterases, IRs, SNMPs, and iGluRs, and does not include discussion of ORs. That said, ORs have been shown to exhibit conserved characteristics across a wide range of species.

Round 2

Reviewer 1 Report

Comments and Suggestions for Authors

I don't have further comments for this revised manuscript. 

Author Response

Thank you to the reviewer for our suggestions!

Reviewer 3 Report

Comments and Suggestions for Authors

Review report for Revised version of:

Manuscript ID: insects-3598958

Title: Mechanisms of the impact of Alnus ferdinandi-coburgii odor substances on host location of Tomicus yunnanensis

Dear Authors,

I am glad to find that you have addressed most of the suggestions and concerns in the revised version of the manuscript. I appreciate the effort and most corrections are accepted. Most of them are considered as improvements and the quality has been improved to acceptable standards. However, I have a comment on the current interpretation of phylogenetic analysis.

L288-305:  It is perfectly fine to use phylogenetic analysis only as a tool for clustering purpose and not for exploring homology related aspects. Given that, the unrooted NJ-trees without outgroups provided as main figure and supplementary figures can stay only if, there is no evolution-related interpretations like orthologyor paralogy. Therefore I strongly suggest authors to remove such terms (orthologs and paralogs) in the two specified paragraphs and use clustering as it refers to.

L289: It is up to the authors to make use of evidence and annotate the co-receptors like ORCo, considering that you have full-length sequence with strong sequence similarity. Unfortunately I cannot access any sequence. It would have been better if authors included sequences in the supplementary table S1. The NCBI dataset is not valid/not accessible at PRJNA1253926 (L:520). Please recheck.

minor:

L398: incomplete (;) ?

Best,

Reviewer.

Author Response

Dear Reviewer:

We sincerely appreciate the reviewers’ valuable comments and suggestions. We have carefully addressed all the concerns and revised the manuscript accordingly. Below are our point-by-point responses, with the changes highlighted in the revised manuscript.

L288-305:  It is perfectly fine to use phylogenetic analysis only as a tool for clustering purpose and not for exploring homology related aspects. Given that, the unrooted NJ-trees without outgroups provided as main figure and supplementary figures can stay only if, there is no evolution-related interpretations like orthologyor paralogy. Therefore I strongly suggest authors to remove such terms (orthologs and paralogs) in the two specified paragraphs and use clustering as it refers to.

Return: Thank you for your valuable comments, this section has been revised.

L289: It is up to the authors to make use of evidence and annotate the co-receptors like ORCo, considering that you have full-length sequence with strong sequence similarity. Unfortunately I cannot access any sequence. It would have been better if authors included sequences in the supplementary table S1. The NCBI dataset is not valid/not accessible at PRJNA1253926 (L:520). Please recheck.

Return: Thank you for your valuable comments, this section has been revised.

L398: incomplete (;) ?

Return: Thank you for your valuable comments, this section has been revised.